# Analyzing Polyethylene Terephthalate Bottle Waste Technology Using an Analytic Hierarchy Process for Developing Countries: A Case Study from Indonesia

**Akhmad Amirudin** [1,2,*]**, Chihiro Inoue** [1] **and Guido Grause** [1,3]

1   International Environmental Leadership Program, Environmental Studies for Advance Society Department, Graduate School of Environmental Studies, Tohoku University, Aramaki Aza Aoba 6-6-20 Aoba-ku, Sendai 980-8579, Japan
2   Department of Public Administration, Faculty of Administrative Science, Brawijaya University, Malang 65145, Indonesia
3   School of Chemical Engineering, University of Birmingham, Edgbaston, Birmingham B15 2TT, UK
*   Correspondence: akhmad.amirudin.r2@dc.tohoku.ac.jp

**Abstract:** PET bottle waste is easy to recycle because it is easy to separate, abundant, and competitively priced. Technologies for the treatment of PET bottle waste have been evaluated to date by using life cycle assessment (LCA), but this does not take into account all of the aspects that policymakers consider necessary when selecting an acceptable technology. Aspects such as society, economics, policies, and technical applicability need to be considered along with the environment and resource consumption to complement the LCA results for PET bottle waste. These aspects were selected as criteria for the analytical hierarchy process (AHP), and stakeholders were invited to make a comparison evaluation of the criteria and sub-criteria. Academics were involved to compare the technology options. The results show that society is the highest priority because it is the main actor that ensures the application of the technology, and that job creation is the most important indicator for the selection of the technology in society criteria. After comparing open landfills, sanitary landfills, incineration with energy recovery, pelletizing, glycolysis, and hydrolysis for the utilization of PET bottle waste, this study suggests pelletizing as the acceptable technology for Indonesia because pelletizing is dominant in all the criteria and sub-criteria which support sustainability in waste management. This is the first time that a single plastic fraction that is easy to collect and recycle has been studied with the AHP. The results show that this type of plastic could also be reused in developing countries through mechanical recycling.

**Keywords:** multicriteria comparison; plastic waste management; technology comparison; stakeholder's judgment; waste bank; decision making

## 1. Introduction

Polyethylene terephthalate (PET) is one of the most widely used polymers for packaging due to its durability, transparency, and gas barrier properties [1]. It is expected that about 583 billion plastic drinking bottles will have been produced in 2021 [2], of which 62% will have been made from PET [3]. Discarded PET, on the other hand, is difficult to degrade under environmental conditions and releases toxins into soil and water, as well as emissions into the air, which are harmful to humans and biodiversity [4]. The two most widely used technologies for PET waste management are landfilling and incineration [4,5], but both of these methods have negative environmental impacts. For example, landfilling PET produces volatile organic compounds, such as xylenes, ethylbenzenes, and trimethylbenzenes [5], while incineration releases toxic gases and carbon dioxide.

In Indonesia, 69% of waste is treated in open landfills. The rest is traditionally buried, composted, or openly burned; about 9% remains untreated [6]. There are two national

municipal solid waste management frameworks (Act No. 18/2008 and Law No. 81/2012). The fact that both regulate the treatment of plastic waste (more specifically, PET bottle waste) increases uncertainty in the management of PET bottle waste and reduces the quantity and quality of PET waste available for recycling. In Indonesia, there are still many landfill applications. This leads to the need to prioritize applications that can reduce the burden of environmental damage, are affordable for investors, and are supported by the community.

To set the research framework, this study elaborated on and discussed Indonesian waste management and PET bottle disposal technology. It is pointed out that Indonesia has the second-largest amount of poorly managed waste in the world, which is one of the major unsolved environmental problems [7,8]. Moreover, PET bottle waste is one of the most promising sources for elaborating on waste-to-treasure concepts [3]. Furthermore, selected technologies for municipal solid waste management are also considered suitable for PET bottle waste valorization.

PET bottle waste recycling contributes to the circular economy by reducing the extraction of new materials and environmental pollution, thereby increasing sustainability [9]. By selecting a suitable technology, PET bottle waste can be transferred from an open-loop process to a closed-loop one. In the research on plastic waste recycling, the comparison of technologies becomes important, as much of the technological development is completed in the laboratory stage [10]. However, implementation at the policy level is still minimal, as can be seen from stakeholder responses in the area of plastic recycling, especially PET bottle waste.

The recycling of waste is an important strategy for developing countries because recycling can reduce environmental pollution and dependence on resource exploitation, and also improve economic growth and waste management [11,12]. In order to utilize waste after recycling, it is very important to determine the appropriate waste treatment technology, because waste technologies require not only facilities and equipment, but also stakeholder dialogue and public participation as well as acceptance [13,14]. These require studies on PET technology selection in developing countries [4,15,16].

The AHP can be used to facilitate dialogue among stakeholders to find an appropriate technology. This method is able to evaluate different criteria and aspects to reach an equal justification and assessment by involving stakeholders and experts in the field of PET bottle waste management. This method has been used in several studies on technology selection at the national level [9,16,17], but PET bottle waste has not been widely considered. Other research comparing PET bottle waste technologies using multicriteria decision making (MCDM) has not been conducted to date. Although life cycle assessment (LCA) studies have been conducted to evaluate the environmental impacts of different treatments, they do not provide information on their feasibility or acceptability by stakeholders and the public. The contribution of this study is to fill this gap and evaluate the criteria and subcriteria to be considered when selecting an acceptable technology for recycling PET bottle waste.

However, it is still difficult for policymakers to determine the appropriate technology for PET bottle waste treatment when considering economic, social, and environmental aspects in Indonesia. Therefore, this study compares criteria and sub-criteria in terms of their relevance for future technology in Indonesia and evaluates alternatives for PET bottle waste treatment in terms of their acceptability by stakeholders.

## 2. Literature Review

The selection of waste technology has been discussed in several studies, which differ in terms of their targets, regions, methodologies, waste types, and outcomes. Yap and Nixon (2015) [18] searched for the most effective technology for energy recovery. Asefi and Lim (2017) [19] established an integrated solid waste management model that considered economic, social, and environmental benefits. Other studies addressed the selection of the most appropriate technology to maximize electricity generation [20]. Waste technology selection studies have been conducted in many countries and regions, such as China [21],

Nigeria [22], Indonesia [23], Turkey [20], Iran [19], Japan [24], the United States [25], and Mexico [17]. Yap and Nixon (2015) [18] made a comparison between India and the United Kingdom.

Various methods and techniques were used to evaluate technology selection in waste management and provide various results and suggestions for improving waste management. Researchers analyzed waste technology selection, focusing mainly on municipal solid waste problems [26,27]. A few researchers focused on specific wastes, such as infectious medical waste, electronic waste, and plastic waste [9,28,29]. However, PET bottle waste has not yet been studied, although PET bottle production is increasing and PET bottle waste has potential for recycling.

Life cycle assessment (LCA) is one method that compares the treatments of PET bottle waste. Bałazińska et al. (2021) [30] compared systems for PET bottle waste, such as recycling, energy recovery, and disposal, without explicitly mentioning the technology used. The result was that recycling had the least negative impact on the environment. Damayanti and Wu (2021) [31] compared the effectiveness of different technological processes for PET plastic waste in mechanical and chemical recycling. They concluded that mechanical recycling leads to the downcycling of materials, while chemical recycling requires the processing of very clean PET waste. Finally, Foolmaun and Ramjeawon (2013) [32] used life cycle sustainability assessments (LCSAs) to compare four disposal scenarios with different levels of landfilling and concluded that a 4–75% flake and 25% landfill fraction are best. Life cycle assessment for plastic waste in Indonesia was observed by Neo et al. (2021) [33], who concluded that open burning significantly contributed to climate change and that landfills are the main contributors to marine ecotoxicity, both of which are two schemas that are quite dominant in Indonesia. Their research recommends starting investment in mechanical recycling for plastic bottle waste, but choosing the right technology requires the consideration of various other relevant aspects and criteria.

Using the AHP, technology selection was performed for different waste types, locations, and goals. Kurbatova and Abu-Qdais (2020) [16] compared landfill gas, anaerobic digestion, incineration, and refuse-derived fuel production, using environmental, health, technical, and socioeconomic criteria for municipal solid waste to select waste-to-energy options in Moscow, Russia. Qazi et al. (2018) [34] used this method to select the right technology for energy recovery from municipal solid waste in Oman. Eight alternative technologies (incineration, gasification, pyrolysis, plasma arc gasification, thermal depolymerization, hydrothermal carbonization, anaerobic digestion, and fermentation) were compared based on five criteria (waste quality and quantity, economic, environmental, technical, and social acceptability). Voudrias (2016) [35] selected incineration, steam disinfection, microwave disinfection, reverse polymerization, and chemical disinfection for infectious medical waste based on environmental, economic, technical, and social criteria in Greece. Azahari et al. [36] studied the treatment of municipal solid waste using the AHP and concluded that a combination of recycling and composting was the best option for Kelantan (Malaysia). Delvere et al. [37] did not reach a clear conclusion for the best way to recycle fiber-reinforced plastic. Finally, there are a few examples of selecting methods for treating plastic waste using the AHP. Although we did not conclusively evaluate the results of these studies because the types of waste and the technologies compared were different, we emphasize that the AHP is still one of the appropriate methods for selecting acceptable and applicable technologies. The literature review summary is presented in Figure 1.

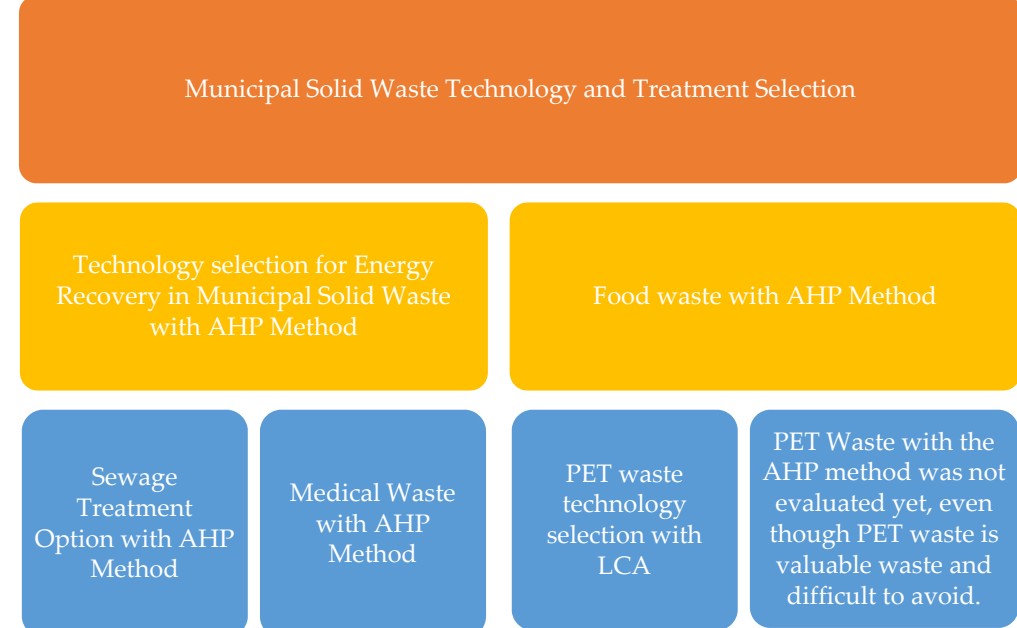

**Figure 1.** Literature review summary.

### 3. Methodology

The analytic hierarchy process (AHP) is a method for decision making by disentangling the hierarchy into goals, criteria, and alternatives [38]. This process develops preference options for policymakers and structures information from complex situations [39]. Therefore, the AHP is used to analyze material recycling and resource utilization applications by comparing inventory and demand, and helps policymakers to set priorities based on the AHP results [40,41]. It is used to develop preference flows and inventories for plastic waste and creates systems to ensure that plastic waste can be converted into valuable resources.

#### 3.1. Goal and Criteria Identification

PET bottles are still difficult to replace and avoid in Indonesia; there is no special post-consumption treatment, so the purpose of this study is to select the most acceptable technology for PET bottle waste utilization in Indonesia. To achieve this goal, the AHP is used to determine criteria to be used as a reference for evaluating alternative technologies. In addition to these criteria, sub-criteria are also formulated to be used as indicators for determining the technology for PET bottle waste.

Figure 2 shows the hierarchy used to achieve the goal of recycling PET bottle waste. The selected criteria and sub-criteria represent important points to be considered in determining the technology, and complement LCA, whose focus is not the calculation of environmental indicators. These criteria and sub-criteria using a questionnaire with a paired comparison of stakeholders and experts, which allows a ranking of the criteria to be calculated (the questionnaire can be found in the Supplementary Material). The main criteria and sub-criteria were identified through extensive literature research. The criteria are environment, resource consumption, society, economy, policy, and technical applicability [22,42–45]. A more detailed definition of the sub-criteria can be found in Table 1.

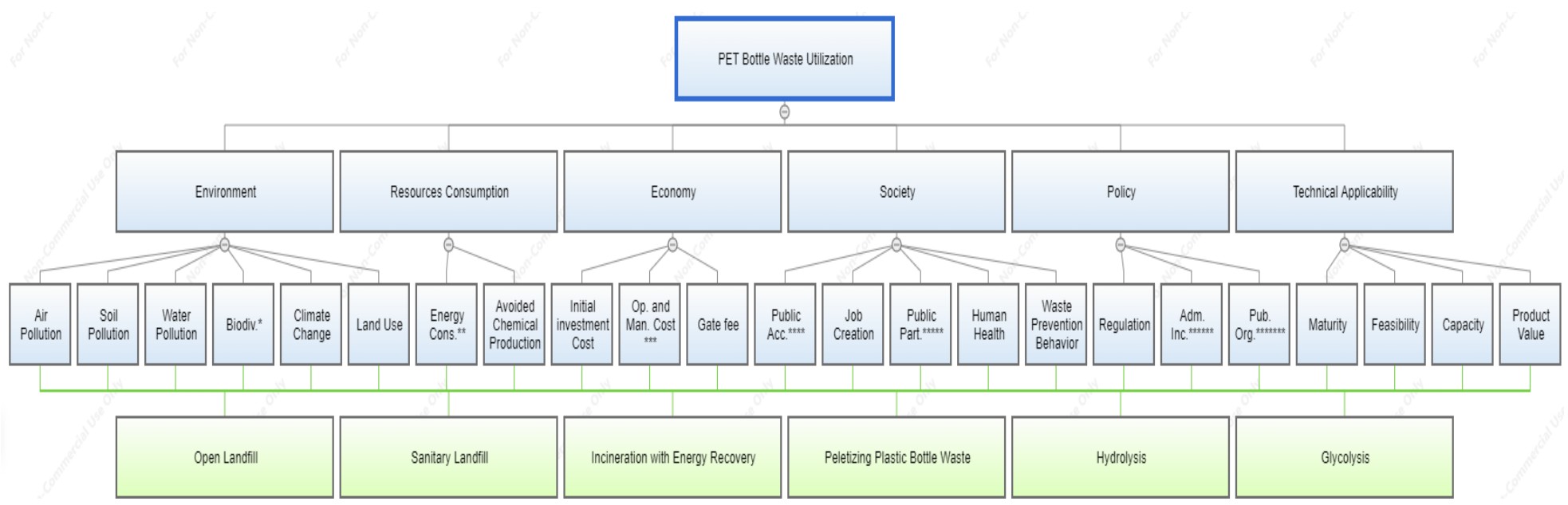

**Figure 2.** Analytical hierarchy construction. Annotations: * biodiversity; ** energy consumption; *** operation and management cost; **** public acceptance; ***** public participation; ****** administrative incentive; and ******* public organization.

**Table 1.** Criteria and sub-criteria definition.

| Criteria | Sub-Criteria | Reference |
|---|---|---|
| Environment | Air pollution | [26,46] |
| | Soil pollution | [46] |
| | Water pollution | [46] |
| | Biodiversity | [47] |
| | Climate change | [47] |
| | Land use | [21] |
| Resource consumption | Energy consumption | [48] |
| | Avoided chemical production | [4] |
| Economy | Initial investment cost | [20,22] |
| | Operation and management cost | [16] |
| | Gate fee | [18] |
| Society | Public acceptance | [34,44] |
| | Job creation | [44] |
| | Public participation | [49] |
| | Human health | [4] |
| | Waste prevention behavior | [44,50] |
| Policy | Regulation | [18] |
| | Administrative incentive | [18] |
| | Public organization | [15,43] |
| Technical applicability | Maturity | [22] |
| | Feasibility | [20] |
| | Capacity | [20] |
| | Product value | [51] |

*3.2. Analytical Hierarchy Construction*

The hierarchical structure in the AHP is very important to facilitate the identification of problems, indicators, and proposed solutions. In this study, a hierarchy with 4 levels is used (Figure 2), which consists of the goal, criteria, sub-criteria, and alternatives. The first level describes the goal to be achieved in this study. The goal is to utilize PET bottle waste in Indonesia, which can be achieved by using different alternatives at the fourth level, each of which has advantages and disadvantages (Table 2).

The alternatives offered to stakeholders for selection were chosen based on the degree of the applicability of the technology in Indonesia. In this study, 6 alternatives were selected: 4 of them have already been implemented in Indonesia, such as open landfills, sanitary landfills, incineration with energy recovery, and pelletizing. Glycolysis and hydrolysis were studied by experts from Indonesia, who were also invited to answer the questionnaire in this study.

The alternatives are connected to the goal through the second and third levels. The second level contains the criteria that are important for the evaluation of the different technological alternatives and represent different aspects of the waste technology selection. Due to the complexity of the problem and the number of aspects, it is not possible to find a process that meets all of the desirable conditions. For example, the mechanical recycling of PET has the lowest environmental impact but requires community support for waste separation and a purchaser for the recycled material. It can be difficult to find an alternative that offers benefits on all of the criteria examined, and it is necessary to apply weightings

for prioritization according to these criteria. The third level subcriteria are indicators used to define the strengths and weaknesses of each alternative. This is important to facilitate logical decision making to achieve the goals.

**Table 2.** Alternatives: processes for the treatment of PET bottle waste.

| Alternatives | Definition | Source |
|---|---|---|
| Open landfill | Final disposal of untreated waste on separate or excavated areas | [52] |
| Sanitary landfill | Final disposal of waste on excavated areas for different types of waste, covered with soil to reduce the negative impacts; possibility of energy generation from exhaust gases | [53] |
| Incineration with energy recovery | Conversion of waste into energy (electricity and heat) | [44] |
| Pelletizing plastic bottle waste | Remelting and extrusion of PET to be used as a raw material | [10] |
| Hydrolysis | Conversion of PET at high temperatures and pressures to produce terephthalic acid and ethylene glycol | [10] |
| Glycolysis | Conversion of PET into ethylene glycol to produce bis(2-hydroxyethyl)terephthalate (BHET) | [10] |

*3.3. Pairwise and Stakeholder Opinion*

Stakeholder opinion and judgment is an essential step in the AHP. A questionnaire was created to obtain stakeholders' opinions and judgments using a scale comparison, as described in Table 3.

**Table 3.** Saaty's fundamental scale.

| Score | Definition |
|---|---|
| 1 | Equal importance between two criteria |
| 2 | Between equal and weak difference between criteria |
| 3 | Weak difference between criteria |
| 4 | Between weak and strong difference between criteria |
| 5 | Strong difference between criteria |
| 6 | Between strong and demonstrated difference between criteria |
| 7 | Demonstrated difference between criteria |
| 8 | Between demonstrated and absolute difference between criteria |
| 9 | Absolute difference between criteria |

Adapted from [16,38].

The stakeholders in the AHP are important to make a comparative judgment of the criteria and sub-criteria, determine the score that indicates how important a criterion is to achieving the goal, as well as compare all elements in pairs. The pairwise comparison in the form of a matrix gives the priority. The rating scale ranges from 1, which means that both elements are equally important, to 9, which means the greatest discrepancy between the importance of the two elements. Brainstorming sessions, in-depth discussions, and the questionnaire are used by the stakeholders to improve the comparative assessment.

Stakeholders were selected based on their knowledge of plastic waste and their position in a particular organization, such that their opinion can represent the conditions in Indonesia. Although subjectivity cannot be completely eliminated, each stakeholder is a person who knows the problems of plastic waste in Indonesia. The stakeholders were selected from different areas of public life related to waste management, especially plastic waste, i.e., leaders of ministerial departments responsible for waste management and representatives of waste banks as well as plastic bottle manufacturers. As shown in

Tables 4 and 5, for each category of stakeholders and experts, 2 to 4 were selected. The number of different stakeholders and experts was limited by their availability and willingness to participate in this research.

**Table 4.** Stakeholders' categories.

| No. | Stakeholder's Category | Description | Number |
|---|---|---|---|
| 1. | Government | Head department of waste policy and management | 3 |
| 2. | Mineral water producer and seller | Prominent producer and seller of mineral drinking water | 2 |
| 3. | Recycler industry association | Head of the association of the waste plastic recycling industry | 2 |
| 4. | Scavenger association | Head of the scavenger and waste picker association | 3 |
| 5. | Nongovernmental organization | Nongovernmental organizations focused on environmental and waste problems | 4 |
| 6. | Public opinion leader | Prominent personality actively addressing the plastic waste problem | 3 |
| 7. | Waste bank operator | Community organization operator for sorting municipal solid waste | 4 |
| 8. | Household | Households joining plastic waste utilization groups | 3 |
| | | **Total** | **24** |

**Table 5.** Expert category.

| No. | Expert Category | Description | Number |
|---|---|---|---|
| 1. | International expert | International scholars specialized in the research on waste technology | 4 |
| 2. | Indonesian expert | Indonesian professors of integrated solid waste treatment technologies | 3 |
| | | **Total** | **7** |

The experts were selected through search engines and platforms, such as ResearchGate, Scopus, Web of Science, and Google Scholar, to determine if they had experience in the fields of plastic waste and problems related to plastic waste management. Experts were selected for the evaluation of the alternatives by using the search terms "polyethylene terephthalate" and "waste management" in Scopus. Publications from the last 5 years were ranked by the number of citations. The corresponding authors were asked to participate in this study. Out of 24 invited experts, 7 responded positively.

*3.4. Data Calculation*

Decisions are based on priority weights composed of objectives and sub-criteria, as well as scores collected through a questionnaire sent to stakeholders and experts. The priority of criteria and alternatives is determined by making pairwise comparisons, i.e., criteria, sub-criteria, and alternatives are each compared in pairs in a matrix, as shown in Table 6.

**Table 6.** Pairwise comparison matrix.

| | $A_1$ | $A_2$ | ... ... | $A_n$ |
|---|---|---|---|---|
| $A_1$ | $w_1/w_1$ | $w_1/w_2$ | ... ... | $w_1/w_n$ |
| $A_2$ | $w_2/w_1$ | $w_2/w_2$ | ... ... | $w_2/w_n$ |
| ... ... | ... ... | ... ... | ... ... | ... ... |
| $A_n$ | $w_n/w_1$ | $w_n/w_2$ | ... ... | $w_n/w_n$ |

Adopted from Saaty, T.L. (1990) [54].

The relations $w_n/w_n$ represent the score of Saaty's fundamental scale (Table 3) taken from the questionnaires answered by stakeholders and experts.



The data were processed by Expert Choice version 6.2.001.42753. For each matrix, a consistency check was performed by calculating the consistency ratio (CR) in three steps: First, the eigenvalue ($\lambda_{Max}$) was calculated according to Equation (1):

$$A.w = \lambda_{Max}.w \tag{1}$$

where A is the comparison matrix, w is the normalized eigenvector (priority vector), and $\lambda_{Max}$ is the eigenvalue. When evaluating each comparison, the next step is to arrange the priorities, eigenvalues, and eigenvectors to determine the respective value in the matrix; the total matrix value in each column is compared to the matrix value and summed for each row. Then, the total row values of the calculated matrix are summed. To determine the priority value, the total value of the rows in the matrix is compared to the total value of the calculated column. The eigenvalue is the total number of multiplications of the priority values in the matrix compared to the priority values. The eigenvalue is the sum of the eigenvalues divided by the order of the matrix, n. The detailed result of the calculation is shown in the Supplementary Material.

Second, the consistency index (CI) was calculated according to Equation (2) [16]:

$$CI = \frac{\lambda_{Max-n}}{n-1} \tag{2}$$

Finally, the CR was calculated from the CI and the random index (RI) according to Equation (3) [45]:

$$CR = \frac{CI}{RI} \tag{3}$$

The RI depends on the order of the matrix. The values of Saaty's RI calculation are shown in Table 7. This shows that the result of the CR of this study is 0.0767, which is less than 0.1, which means that the judgments of this study are considered consistent.

**Table 7.** Saaty's random index [38].

| $n$ | 1 | 2 | 3 | 4 | 5 | 6 | 7 | 8 | 9 | 10 |
|-----|------|------|------|------|------|------|------|------|------|------|
| **RI** | 0.00 | 0.00 | 0.52 | 0.89 | 1.11 | 1.25 | 1.35 | 1.40 | 1.45 | 1.49 |

In this study, technology for the utilization of PET waste was considered and evaluated to obtain a technology proposal that has the highest weight to serve as a solution for PET bottle waste utilization based on the given criteria and sub-criteria. Six criteria with twenty-three sub-criteria (Table 1) were evaluated by stakeholders in Indonesia (Table 4), and six technology alternatives were compared by experts (Table 5) based on an online survey and literature review related to this research. Expert Choice version 6.2.001.42753 was used to calculate the priority rating.

### 3.5. Sensitivity Analysis

A sensitivity analysis was used to determine how the change in the priority of the criteria affects the order of the alternatives. This is an integral part of any decision-making process accompanied by the creation of a decision support model [55]. The sensitivity analysis was performed using Expert Choice. In this work, the results of the expert judgment were compared to a scenario in which all criteria had the same priority of 16.7%.

### 4. Analytical Hierarchy Results

This study compares six alternatives to utilize PET bottle waste (open landfills, sanitary landfills, incineration with energy recovery, pelletizing plastic bottle waste, hydrolysis, and glycolysis) by six criteria (environment, resource consumption, economy, society, policy, and technical applicability) with twenty-three sub-criteria. After all of the questionnaires were received from the stakeholders, the results of the questionnaires were transferred

to the web-based version of Expert Choice, the AHP application used in this work. Two different questionnaires were developed: First, all of the stakeholders, with the exception of the international experts, were comparing the criteria and sub-criteria questionnaire. Second, the international and Indonesian experts compared the alternatives via the criteria and sub-criteria.

The pairwise comparison between the criteria to achieve the priority vector is shown in Table 8. The value of each criterion was the average of the stakeholders' comparative judgments. The pairwise comparison matrix for the sub-criteria is shown in Table S2 of the Supplementary Material.

**Table 8.** Pairwise comparison matrix.

|  | Environment | Resource Consumption | Economy | Society | Policy | Technical Applicability | Priority Vector |
|---|---|---|---|---|---|---|---|
| **Environment** | 1 | 1 | 3 | 0.5 | 3 | 3 | **0.227** |
| **Resource consumption** | 1 | 1 | 1 | 0.5 | 2 | 2 | **0.167** |
| **Economy** | 0.33 | 1 | 1 | 0.33 | 0.5 | 0.5 | **0.086** |
| **Society** | 2 | 2 | 3 | 1 | 3 | 2 | **0.298** |
| **Policy** | 0.33 | 0.5 | 2 | 0.33 | 1 | 2 | **0.119** |
| **Technical applicability** | 0.33 | 0.5 | 2 | 0.5 | 0.5 | 1 | **0.103** |

*4.1. Criteria and Sub-Criteria Comparison*

The first result was the comparison of criteria and sub-criteria for the evaluation of PET bottle waste treatment technologies in terms of their relevance (Figures 3 and 4), the data for which is presented in Table 8. The selection of impact criteria and subcriteria indicators must be consistent with the objective of the study. The stakeholder panel chose social impact (29.8%) as the most important criterion for PET bottle waste management, followed by the environment with 22.7%. Resource consumption, policy, and technical applicability contributed 16.7%, 11.9%, and 10.3%, respectively. Economic aspects were ranked as the least important criterion, with about 8.6%.

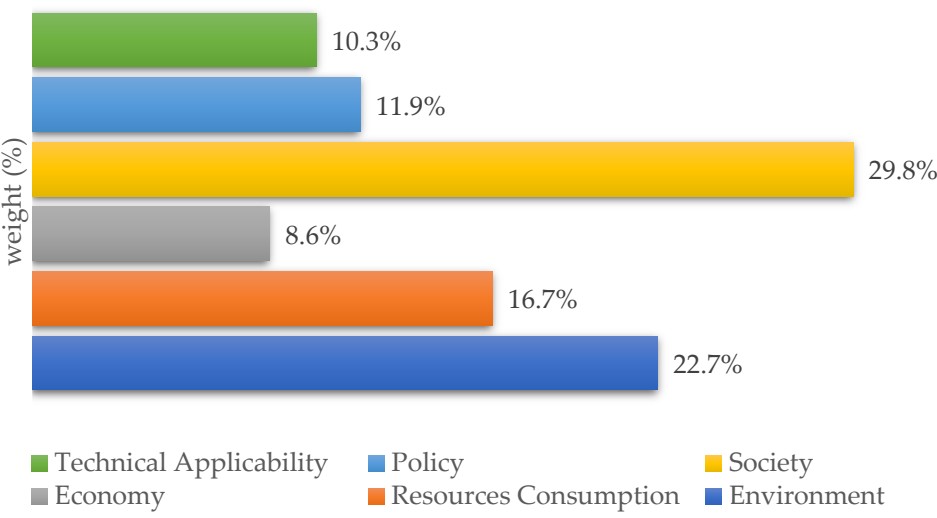

**Figure 3.** Criteria priority (%).

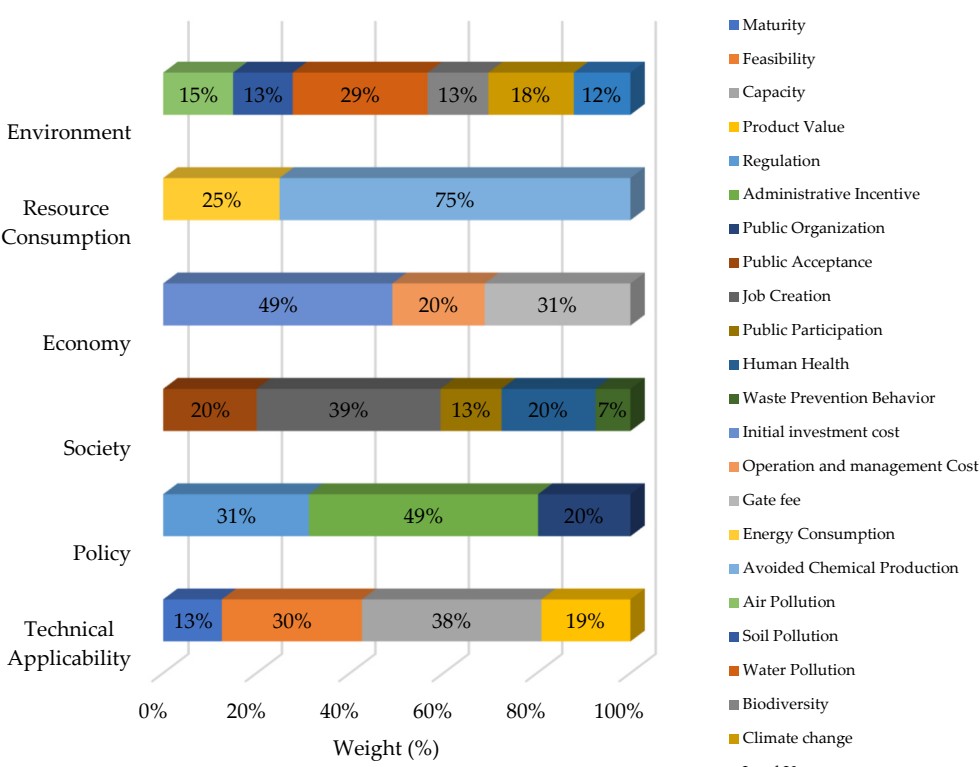

**Figure 4.** Sub-criteria priority (%).

Society (29.8%) was chosen as the most important aspect to consider when defining the technology for PET bottle waste, as communities directly contribute to and support the application of the technology. PET bottle waste is municipal solid waste generated by daily consumption, and needs to be sorted before disposal so that it can be processed with appropriate technology. Therefore, the community plays an important role in waste collection as an important helper in the circular processing of PET bottle waste [56].

The results of this study indicate that all stakeholders agree that the community must be used as the most important factor in determining the technology for plastic bottle waste treatment. This is due to low community participation and awareness in waste management in Indonesia [57]. In Japan, sorting waste, such as paper, cans, and PET bottles, has been shown to be very effective in increasing recycling [58]. Community input in managing plastic bottle waste increases employment opportunities, impacts health awareness, and reduces waste generation depending on the level of participation and the acceptance of technology and systems, which are generally determined by the government.

Job creation (39%) was the most important indicator for the social aspect. Phillipp and Burdett (2011) [59] estimated that there are 15 million jobs in recycling and waste management activities, and Nurbaiti (2021) [60] noted that about 3.7 million people in Indonesia are employed as waste pickers. Beiler et al. (2020) [61] concluded that job creation in recycling management for packaging waste could be increased and that this should be considered in the waste management system. In Indonesia, waste pickers play a central role in the recycling system [62]. Waste pickers have few alternative ways to earn a living. The shift from an informal to a formal waste management system could lead to unemployment if the informal workforce is not included in future solutions.

Human health (20%) and public acceptance (20%) were the next most important indicators for society. PET bottle waste can endanger human health if PET bottles are contaminated or mishandled during waste management. In developing countries with communities that lack human health awareness, it is very likely that PET bottles are reused for inappropriate purposes that do not meet health standards, for example, reusing PET bottles for inappropriate materials, leaving contaminants in the bottle [63]. Without specific

regulations and mechanisms, recycling PET bottle waste could increase the possibility of health risks from recycled PET bottles to Indonesian society. Public acceptance is also important to ensure community support for separation, collection, and recycling that support the specified technology so that technology is easier to implement and maximizes results according to expectations.

The environment (22.7%) was the second most important aspect in the selection of technology options for PET bottle waste treatment. Air, soil, and water pollution contributed more than 50% to the environmental criterion. Water pollution (29%) was especially important because Indonesia is classified as a country with a high potential for water scarcity in the future [64], while climate change (28%) was selected because Indonesia is vulnerable to sea level rise, droughts, and floods [65]. Climate change encompasses several aspects of human behavior, including production and consumption. This indicates that climate change awareness needs to be considered in the future management of PET bottle waste.

The resource consumption aspect (16.7%) was dominated by avoided chemical production (75%). The chemical process of PET polymerization requires the condensation of terephthalic acid with ethylene glycol [66]. This reaction can be avoided if PET bottle waste is mechanically recycled. The production of terephthalic acid and ethylene glycol can be reduced by recovering monomers through hydrolysis or glycolysis. All these processes support the circular economy. The Indonesian government also recommends that manufacturers minimize the use of virgin materials to produce PET bottles and promotes incentives for recycling (Ministry of Environment and Forestry, 2020) [67].

Policy (11.9%) and technical applicability (10.3%) received moderate ratings from the stakeholder panel. The policy and technical applicability aspects were ranked second highest by government and Indonesian experts, respectively. Other studies also gave a lower priority to policy [68] and technical applicability [26] compared to society and the environment.

The least important criterion was the economy (8.6%), although water producers and sellers (Supplementary Materials) considered the economy aspect to be the most important. However, all of the other stakeholders gave a low priority to the economy for PET bottle waste. The lack of economic capacity (investment costs and operational as well as management costs) is still a problem in Indonesia [57], but it could be solved by cooperation between producers and the government with the concept of extended producer responsibility (Ministry of Environment and Forestry, 2020) [67]. Wang et al. (2020) [68] also concluded that developing countries have a problem with operational as well as management costs (49%%) and gate fees (31%). However, it was difficult to compare these with the environment and society because it was difficult to calculate comparable monetary units.

## 4.2. Expert Evaluation of Alternative Technologies

The experts were international and Indonesian academics who compared the sub-criteria of some selected technologies in pairs to understand their advantages and disadvantages. Figure 5 shows the experts' comparison between the technology alternatives in terms of the individual sub-criteria. Pelletizing proved to be the best solution for all of the sub-criteria. The second most favorable alternative depended on the sub-criteria.

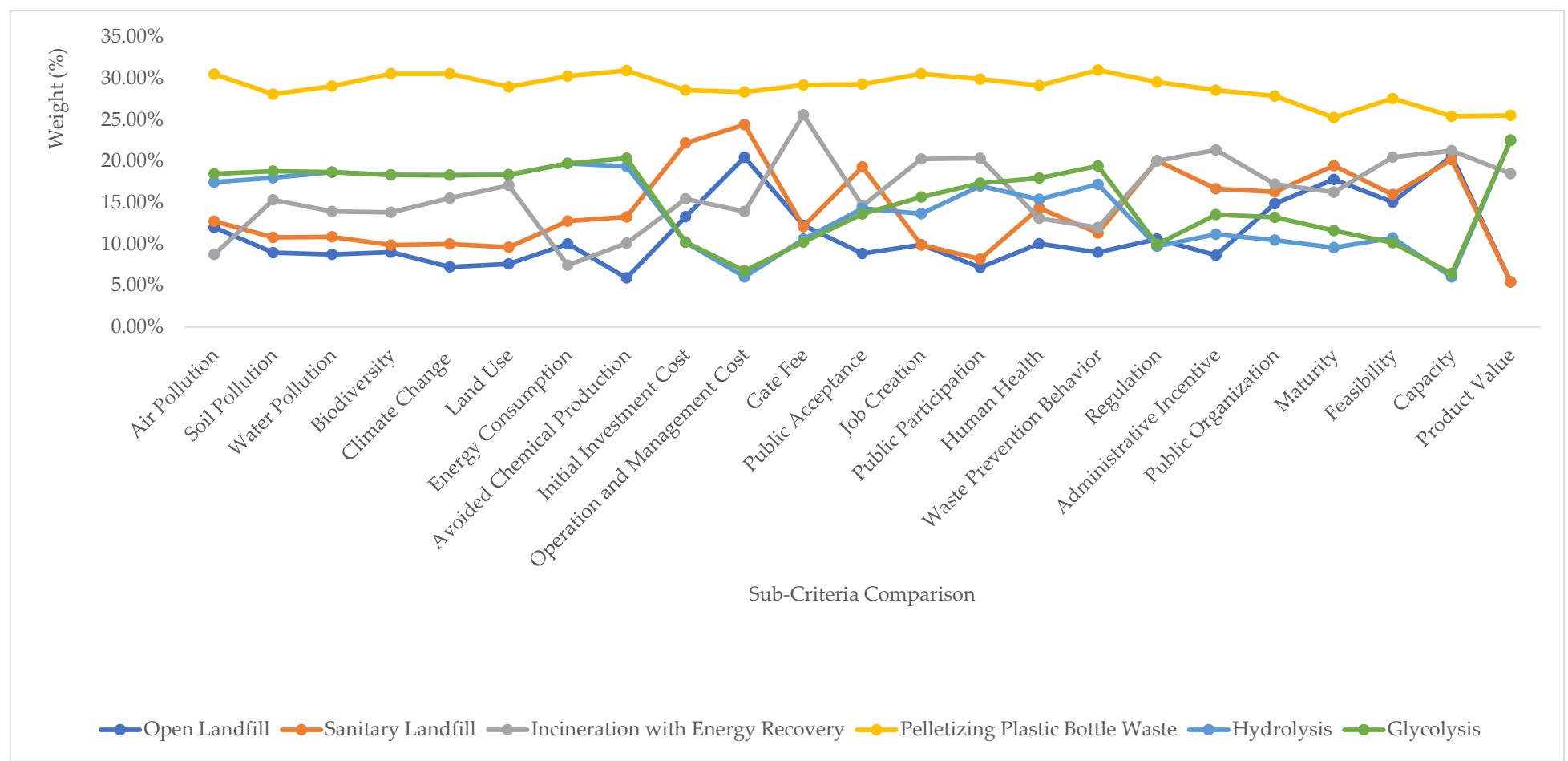

**Figure 5.** Expert sub-criteria judgment results for the alternatives (%).

The ranking of technologies in the environmental subcriteria was very consistent (pelletizing, hydrolysis, glycolysis, incineration, sanitary landfills, and then open landfills) (Figure 5), except for air pollution, where incineration was the lowest in this category. For energy consumption, incineration was the lowest, while for avoided chemical production open and sanitary landfills were the lowest. For hydrolysis and glycolysis, the scores were consistent: second and third place. For the initial cost and operation as well as management costs, hydrolysis and glycolysis ranked the lowest, but at very similar levels.

Open landfills have the lowest public acceptance, at 8.86%, while glycolysis, hydrolysis, and incineration are almost tied, at about 14% (Figure 6). Sanitary landfills are preferred by the population. Job creation and public participation are ranked equally. In addition, hydrolysis, glycolysis, and open landfills are considered to require more complex regulations to implement. In contrast to chemical processes, incineration, sanitary landfills, and open landfills have more capacity, which is contrary to the product value that makes chemical recycling better than the other three.

The stakeholders emphasized job creation, avoided chemical production, and human health (Figure 6). Pelletizing, incineration, and chemical recycling contributed positively to job creation, while landfilling provided the fewest employment opportunities. Glycolysis and hydrolysis allow for avoiding the production of chemicals, as these technologies convert PET waste into chemicals for virgin materials. Incineration still provides electricity and heat, while all of the effort required to produce PET is lost when the material is landfilled. Each of the proposed technologies affects human health, but open landfills are the one with the greatest impact on human health from pollution.

Although pelletizing was preferred by the majority of the experts for all of the alternatives, some differences between the Indonesian and international experts could be identified. The Indonesian experts saw advantages for the regulation of sanitary landfills and incineration (Figures 7 and 8). The international experts gave a different ranking in terms of maturity, feasibility, capacity, and product value. They noted higher maturity and capacity for open and sanitary landfills. Product value was the highest for hydrolysis and glycolysis (Figure 8). The differences are understandable because both expert groups considered these alternatives in different contexts. For example, open landfills have a large capacity compared to the other alternatives. Nevertheless, the Indonesian government has banned the use of open landfills, so the Indonesian experts preferred other alternatives.

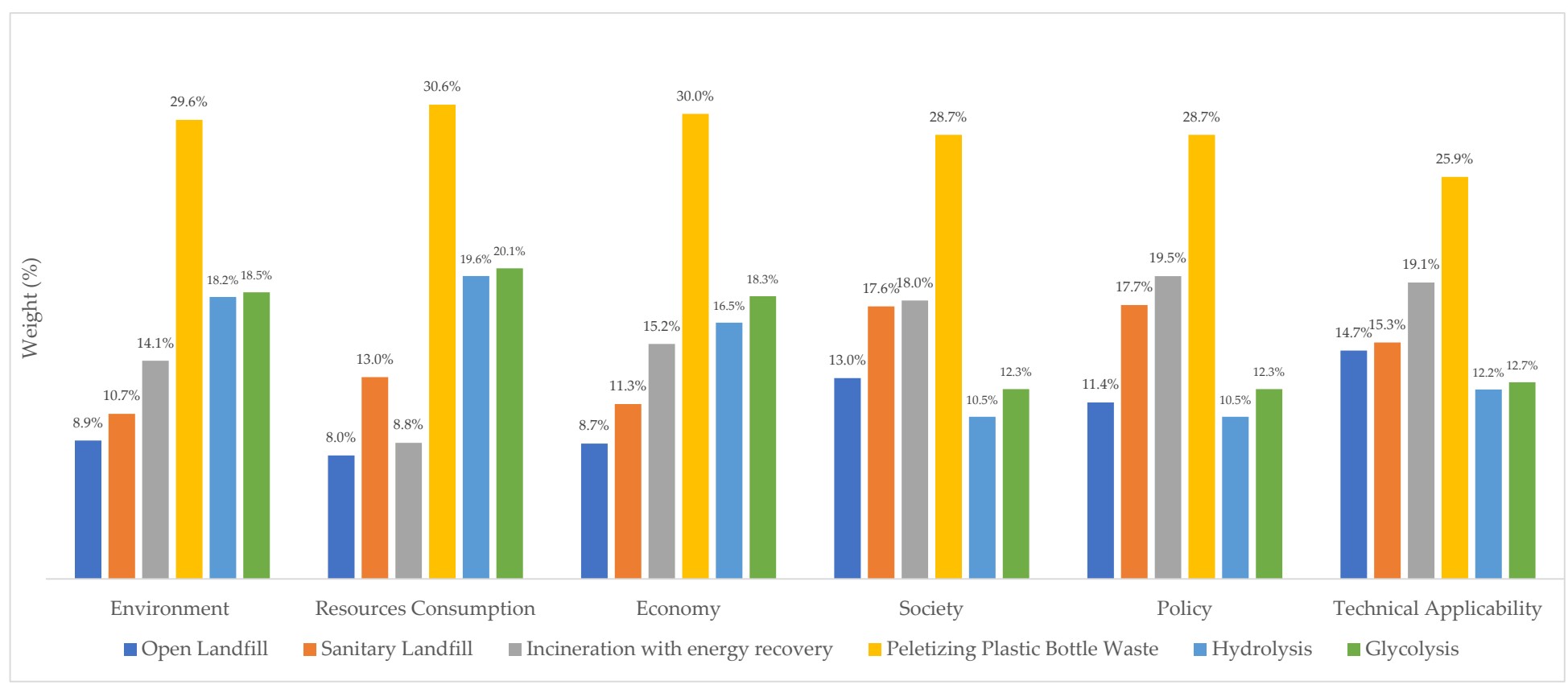

**Figure 6.** Alternatives' weights in criteria (%).

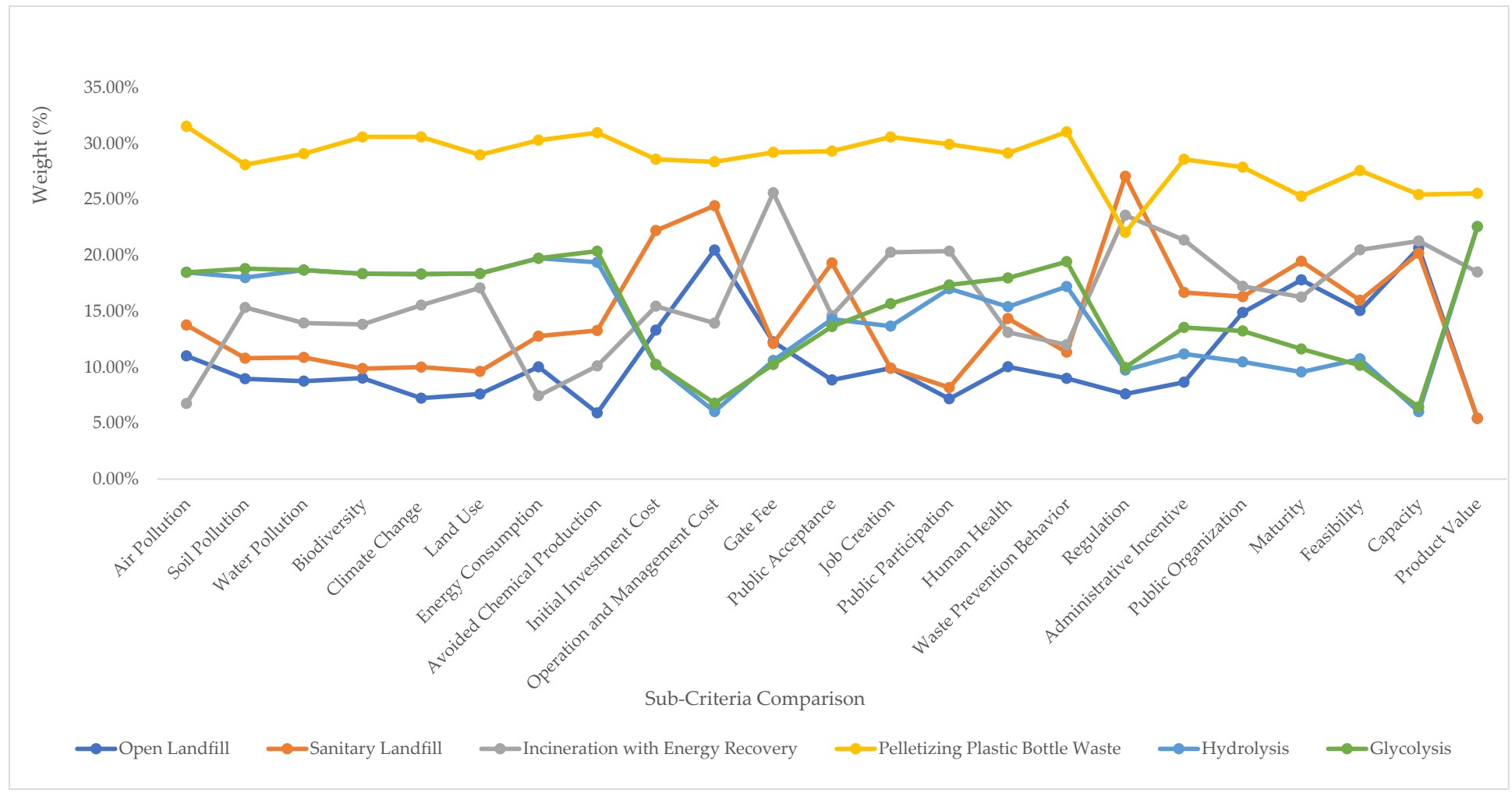

**Figure 7.** Indonesian experts' sub-criteria judgment results for the alternatives (%).

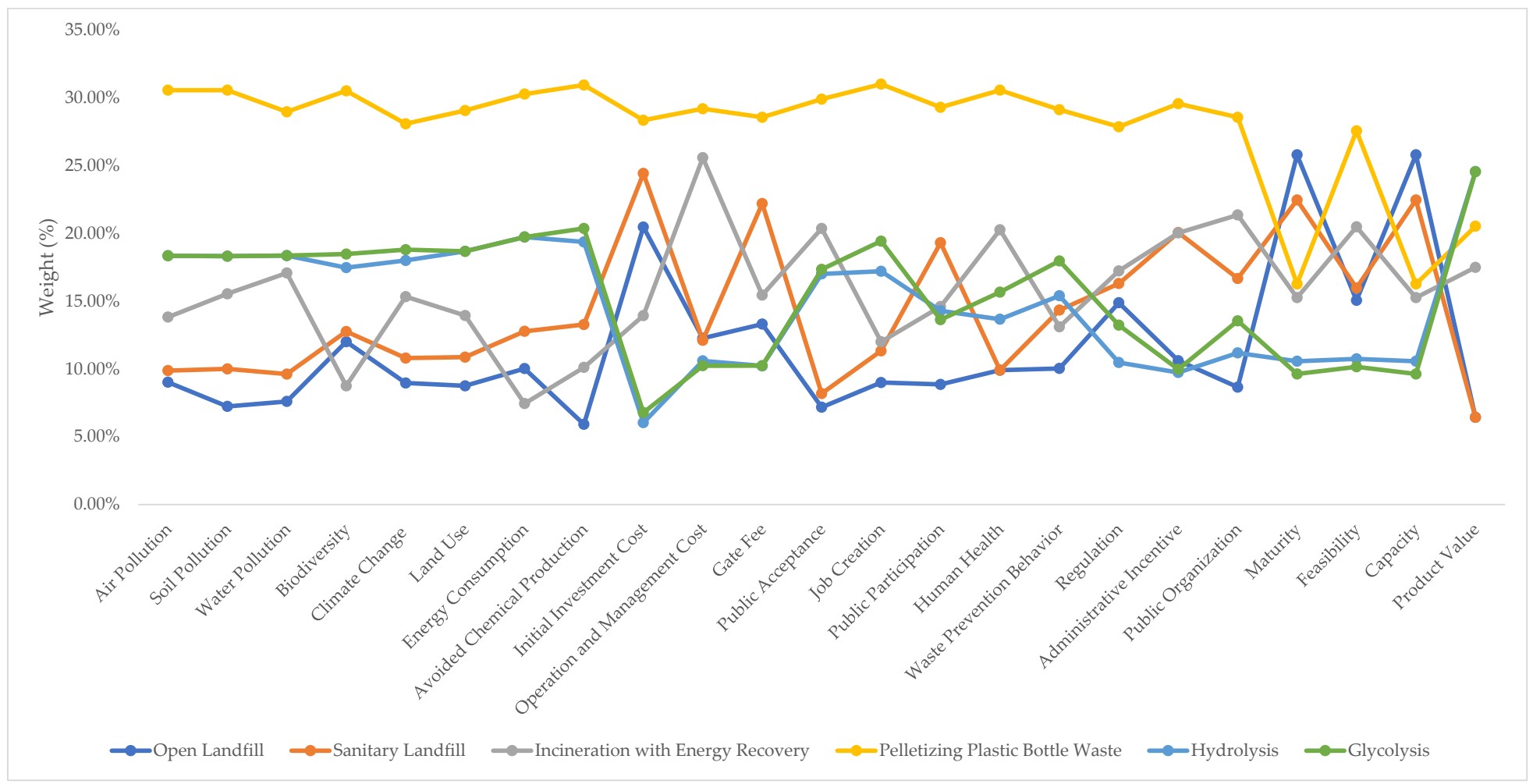

**Figure 8.** International experts' subcriteria judgment results for the alternatives (%).

### 4.3. Comparison of Alternative Technologies

The second outcome of the AHP was to evaluate and propose the acceptable technology for PET bottle waste from a range of alternatives (Figure 9). The pelletizing of plastic bottle waste (28.91%) was the selected alternative for PET bottle waste utilization compared to incineration with energy recovery (16.20%), chemical recycling (glycolysis (15.22%) and hydrolysis (14.47%)), sanitary landfills (14.17%), and open landfills (11.03%).

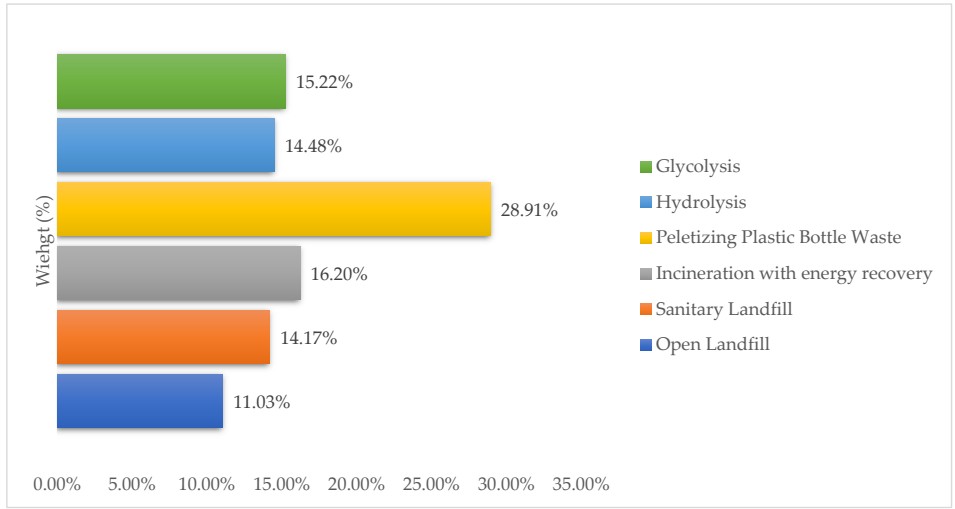

**Figure 9.** Alternatives' priorities (%).

The results for the technology alternatives came from international and Indonesian experts who compared each technology in criteria categories (Figures 5 and 6). Each expert had to compare the alternatives in a specific question to choose the acceptable technology, and PET bottle waste pelletizing was dominant in all of the sub-criteria categories (Figure 6). For the third alternative, there were slight differences between the international and Indonesian experts (Figures 7 and 8). The international experts chose glycolysis as the third alternative, while the Indonesian experts chose sanitary landfills.

Plastic bottle waste pelletizing became an acceptable technology for PET bottle waste utilization as it was superior in all of the criteria. This process was supported by society (28.7%), the environment (29.6%), resource consumption (30.6%), policy (28.8%), the economy (30.0%), and technical applicability (25.9%). Stakeholders felt that plastic bottle waste pelletizing could overcome several indicators in each dominant criterion. For society, the pelletizing of PET bottle waste could be performed by small recyclers and operated by low-skilled personnel in collaboration with waste banks. If this technology was applied in the recycling industry on a small scale, it could create more jobs and increase people's social engagement. Pelletizing could be the best technology for cooperation with waste banks, as it can connect them with industry. This can be beneficial for society and provide a mutual solution to social and environmental problems.

### 4.4. Sensitivity Analysis

Figures 10 and 11 show two AHP models. The first is based on expert judgment, and for the second all of the criteria were set to an equal priority of 16.7%. It can be seen that changes in priority do not significantly affect the ranking of the alternatives. Pelletizing dominates all of the criteria regardless of the priority order, with the economy ranking the highest and technical applicability ranking the lowest. In both models, incineration with energy recovery ranked second and open landfills ranked last. For the alternatives in-between, the positions were shifted somewhat. The experts chose glycolysis and hydrolysis over sanitary landfills, whereas if all criteria had been given equal priority sanitary landfills would have been chosen over glycolysis and hydrolysis. This was only possible because all three of these alternatives were within a range of 1%.

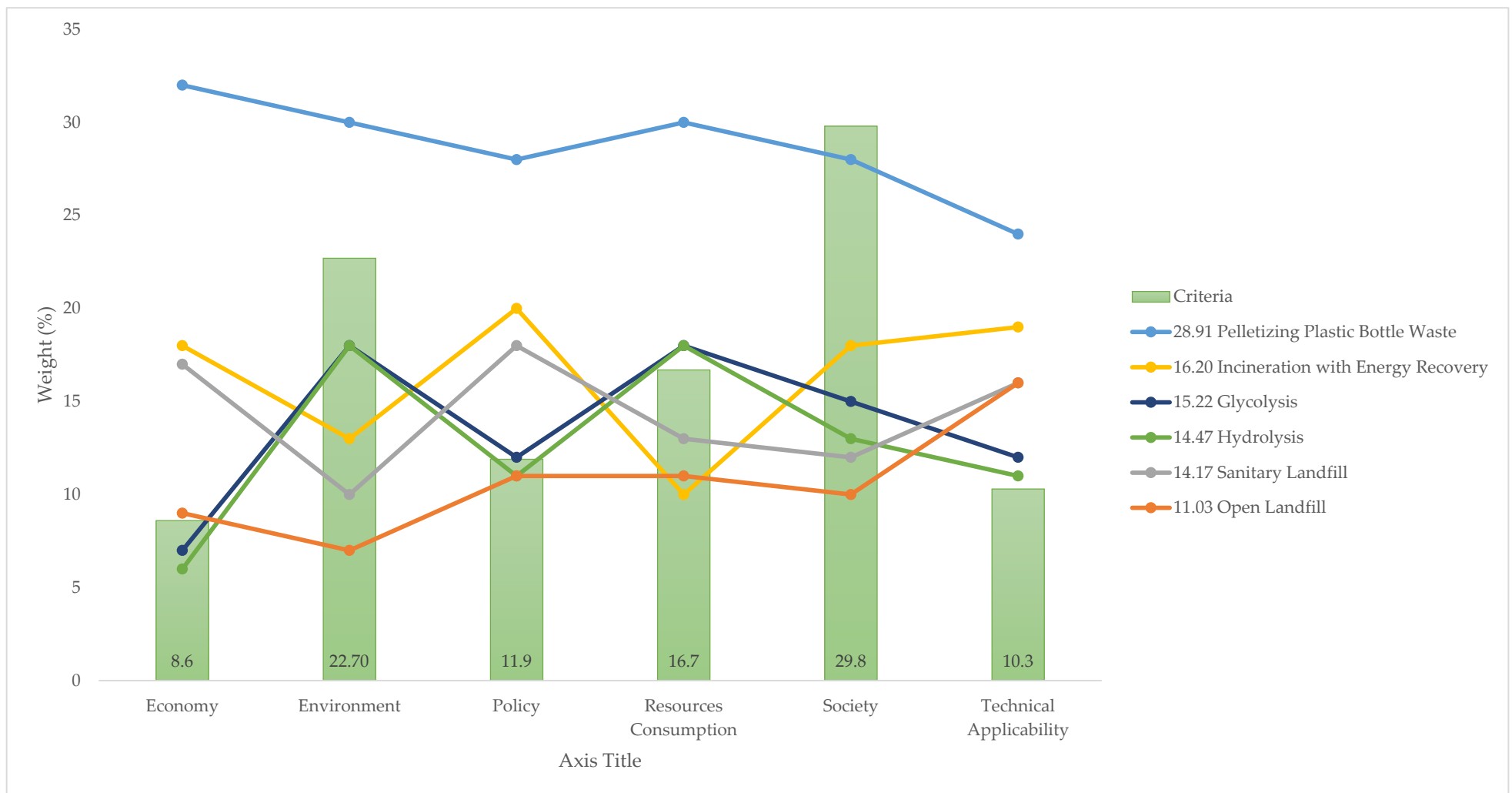

**Figure 10.** Sensitivity analysis: AHP model from experts' judgments (%).

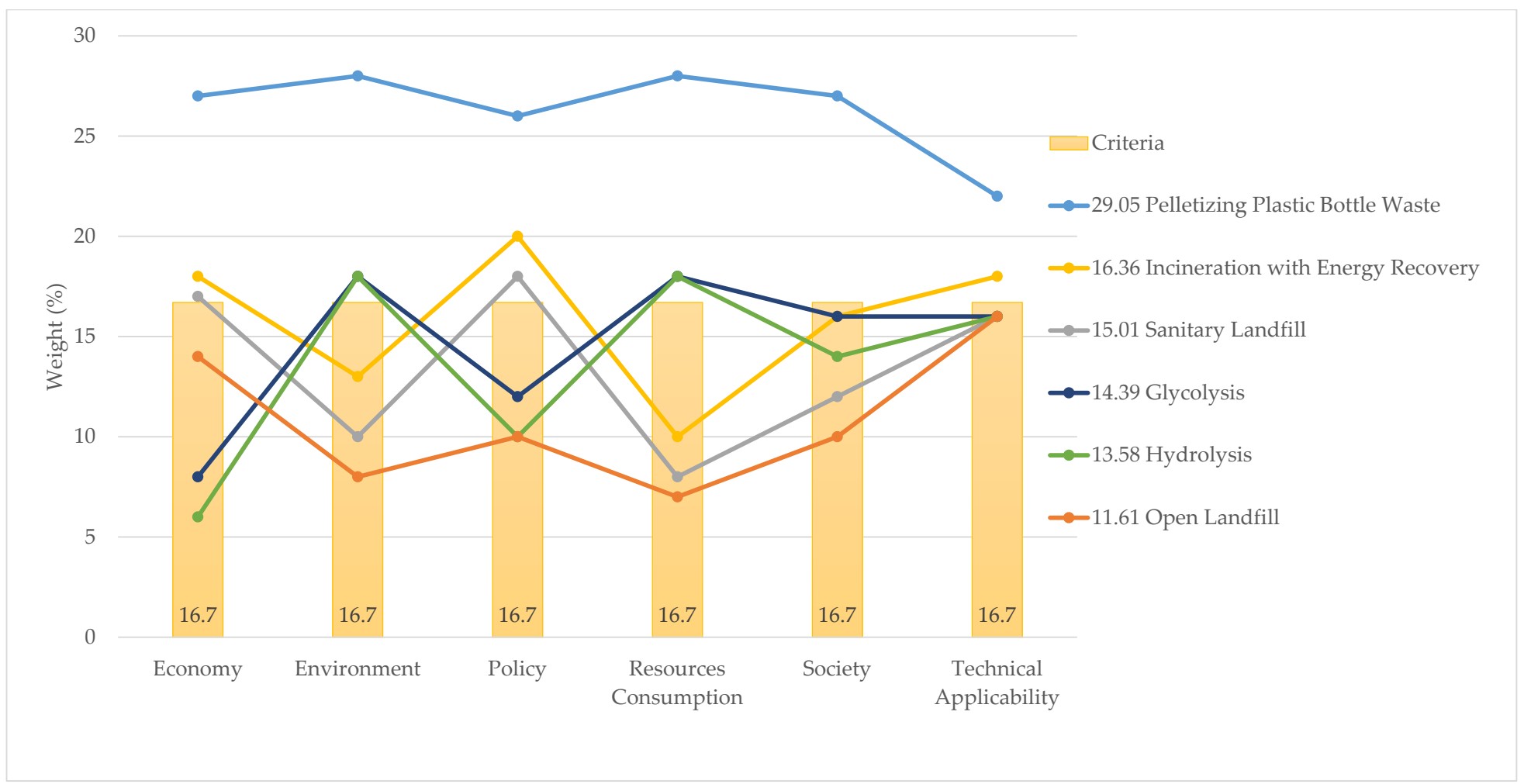

**Figure 11.** Sensitivity analysis: AHP model at an equal priority of criteria (%).

## 5. Discussion

### 5.1. Criteria and Sub-Criteria Implications

Recycling PET bottle waste has many advantages, as it is affordable and can be converted into raw materials or products. However, some developing countries have not yet optimized this advantage. In Indonesia, PET bottle waste has been classified as municipal solid waste, for which there are no specific regulations and technologies for treatment [23,69,70]. Liang et al. (2021) [71] estimated that the generation of plastic waste in Indonesia was 3.01 Mt in 2018, and it continues to increase year by year; most studies estimated that the generation of plastic waste accounts for 12–24% of municipal solid waste [69,72,73]. Sekito et al. (2019) [69] estimated the collection potential for PET bottle waste to be 59 kg/day in Malang City, Indonesia.

Various technologies are used to treat municipal solid waste (including PET bottle waste). Open and sanitary landfills are the predominant technology in Indonesia, and 60% to 70% of municipal solid waste is transported there [74]. Incineration with energy recovery has been introduced in the 12 largest cities selected for the construction of incineration power plants [75]. The pelletizing of PET bottle waste is operated by the waste recycling industry to recycle PET bottle waste into pellets, granulate, or other types of raw materials; an estimated 39% of plastic waste is recycled in Indonesia [33]. Hydrolysis and glycolysis are future options that have not yet been implemented in Indonesia.

The current management of PET bottle waste is open-loop, and some technologies could be used to treat PET bottle waste. PET bottle waste that is not properly collected remains in open landfills, sanitary landfills, or energy recovery incinerators. In the future, collected PET bottle waste could be treated by pelletizing, hydrolysis, or glycolysis. Without the introduction of specific technologies, it would be more difficult to control and minimize the negative impacts of PET bottle waste, as each technology has its own negative aspects.

The results show that society, the environment, and resource consumption are the three most important aspects (Figure 3) that policymakers should consider when selecting an acceptable technology for PET bottle waste. In Indonesia, the contribution of society to waste management is low [76], and the role of scavengers is very important to separate and collect the leftover waste in the plastic waste system [72,77]. This situation encourages stakeholders to make society a priority aspect for PET bottle waste. The role of society is important for the participation in, evaluation of, and contribution to waste management because society is the main actor that makes the system run well [78].

Farahbakhsh and Forghani [79] studied the routing of waste sorting centers using the AHP and set society as a priority aspect because society has the option to support or reject the waste concept or technology. In a study by Foolmaun and Ramjeawon [32] for PET bottle waste in Mauritius, society was ranked as a prominent criterion compared to the environment and the economy because societal criteria could improve job creation as well as social commitment to the technology and therefore increase the success of the implementation of technology. Sandu et al. [78] mentioned that societal categories were environmental NGOs, the plastic industry, local authorities, policymakers, and citizens. Therefore, the stakeholders in this study were included in the previously mentioned societal categories, which is also the reason why society is the main focus in the results of the study.

Environmental aspects were important to consider, as they are the source of the waste problem. The selected technology should reduce the environmental impact; however, each new technology brings its own environmental problems. In addition, environmental issues compete with social, economic, resource consumption, and technical applicability issues for the attention of policymakers, the community, and stakeholders. The outcome varies from country to country depending on priorities. For example, in an evaluation study to select the best scenario for municipal solid waste by Vučijak et al. [80], the emissions to the environment were ranked lower than the income from the sale of the waste. Additionally, in the study by Qazi et al. [34], the environment was not ranked first in the evaluation of waste-to-energy options.

Resource consumption has been used in a few studies to evaluate technology selection, as the trend towards a circular economy has made the discussion of it more familiar. Resource consumption requires the consideration of the contrast between recycling and new production, as recycling also involves resource consumption that may even be higher than resource provision for the production of new materials [81]. Moosavi et al. [82] concluded that energy consumption is the fifth priority and that raw material consumption is the second of the eight total criteria considered in the paper manufacturing process as part of an analytical hierarchy process. In our study, avoided chemical production was in position two of all of the sub-criteria.

### 5.2. Technology Implementation Scenario

The results provide the top three scenarios for discussion (Figures 5 and 6). The first scenario is the application of pelletizing in Indonesia. The expert panel chose pelletizing for the treatment of PET bottle waste in Indonesia. This opens up the possibility of integrating pelletizing into the waste banks, which play an important role in recycling PET bottle waste. Currently, there are 11,603 waste banks in Indonesia, covering almost all provinces and districts [83]. Figures 5 and 6 show that pelletizing scored the highest in all of the criteria categories. It could be an acceptable scenario for PET bottle waste treatment in Indonesia. Pelletizing could be applied in small factories and integrated into waste banks.

The second scenario was incineration with energy recovery. Incineration is less suitable for PET bottle waste because it would be treated in a similar manner to other municipal solid waste. However, 89% of electricity resources in Indonesia are dominated by nonrenewable resources (coal, natural gas, and oil) [84]. Therefore, the government has committed to switch to other energy resources, such as waste and geothermal. Incineration could be accepted as a solution if the separation of PET bottle waste within the collection system is not possible and particularly if contaminated PET bottle waste cannot be reused. Incineration with energy recovery as a second option is supported by the work of Bałazińska et al. [30], who found that the environmental impact of recycling was lower than that of energy recovery, while disposal was the worst scenario. Incineration with energy recovery could be used when electricity demand is difficult to meet through renewable resources. However, Indonesia is still in a position to develop renewable energy generation, such as hydropower and geothermal power.

The third scenario was glycolysis. Glycolysis was the second priority in the use of PET bottle waste from environmental and resource consumption aspects. Glycolysis requires high-quality and clean PET bottle waste [31]. This requirement is difficult to meet in Indonesia because there is no separation process for PET bottle waste. However, with the cooperation of manufacturers, it is possible to collect their PET bottles and send them to glycolysis. Kanchanapiya et al. [85] evaluated the application of glycolysis in Thailand and concluded that it is possible to achieve economic benefits. However, due to the high initial investment costs, collaboration between the private sector and the government is needed to provide funding for the introduction of the technology.

This study suggests that pelletizing should be prioritized in PET bottle waste utilization because pelletizing can be combined with the current treatment of PET bottle waste, in which scavengers and waste banks are the main players in recycling the waste. Pelletizing can be the basis for mechanical recycling at an advanced stage [86], such as improving the quality of fiber outputs, reactive extrusion, which can convert PET waste into filaments, and blending PET with other plastics to obtain more resistant materials, e.g., construction materials and non-food-grade materials. In addition, the government has the ability to support this process through regulations that increase the number of waste banks and the quality of PET bottle waste collected in Indonesia.

A study by Foolmaun and Ramjeawon supports the first and second scenarios [32]. They evaluated PET bottle waste in Mauritius using life cycle sustainability assessment (LSCA). The result was the combination of flake production and landfilling as the first option and the combination of incineration with energy recovery and landfilling as the

second one. Bałazińska et al. [30] compared PET bottle waste management using life cycle assessment (LCA) and recommended that plastic bottle waste should be managed through recycling, as this reduces the environmental impact compared to energy recovery and disposal. This result provides a rationale for the stakeholders' decision in this study, as it shows that social progress and environmental impact reduction are compatible.

AHP results suggest pelletizing as the technology to utilize PET bottle waste in Indonesia, considering that pelletizing has always been number one in 6 criteria and 23 sub–criteria (Figures 5 and 6). AHP emphasizes various backgrounds from stakeholders and experts to propose a consensus that can be considered a solution to environmental problems caused by the management of PET bottle waste. Pelletizing can increase job creation, get administrative incentives from government and capacity to recycle PET bottle waste, reduce water pollution, and avoid chemical production and initial cost investment. Even though pelletizing is a simple technology to convert PET bottle waste to granulate and pellet, for developing countries, it can become an important step for sustainability and a significant level to create bottle-to-bottle recycling.

## 6. Conclusions

In this study, technology options for PET bottle waste utilization were elaborated using the analytical hierarchy process. Stakeholders with different backgrounds and academics were asked to prioritize criteria, sub-criteria, and technology options. The result of the criteria comparison was the order society, environment, resource consumption, technical applicability, policy, and economics. In selecting the technology, society was the most important aspect to consider. The most important sub-criteria were job creation, avoiding chemical production, and human health as the top three. Therefore, the technology must meet these sub-criteria to be acceptable for PET bottle waste.

The most suitable technology was pelletizing, followed by incineration with energy recovery and glycolysis. Pelletizing was the best option because it is easy to apply and could create synergies with waste banks and informal recycling mechanisms in Indonesia to create more jobs. It also reduces material consumption and environmental burden. Given the complexity of the criteria used in this study, this will help policymakers formulate implementation strategies that contribute to suitable waste management and also encourage green investments and financing for the adaption of various new technologies, especially waste treatments in developing countries. However, apart from technology selection, there are other issues that need to be addressed, such as increased community participation, waste collection methods, and industry involvement in PET bottle disposal. In addition, pelletizing is already implemented in some places and meets the infrastructural requirements in Indonesia as many waste banks and local recyclers are already using this technology. There is also social and cultural support, as the Indonesian population collects PET bottle waste to generate additional income. The widespread adoption of pelletizing could have a multiplier effect for many stakeholders.

Since 2019, the Indonesian government has issued several regulations, such as Government Regulation No. 27/2020 on Specific Waste Management, Presidential Regulation No. 83/2018 on Marine Debris Management, and Ministry of Environment and Forestry Regulation No. P.75/2019 on the Roadmap to Waste Reduction by Producers. This shows the commitment and priority of the government in dealing with the waste problem. Therefore, this study can contribute to a better implementation of regulations, especially in the disposal of PET bottles.

For future research on PET bottle waste technology selection, the data required for each sub-criterion should be compared, as this will allow policymakers to make a more detailed calculation of the costs and benefits for each technology. PET bottle waste could become a valuable product that should be managed by the government, waste banks, and the private sector.

**Supplementary Materials:** The following supporting information can be downloaded at: https://www.mdpi.com/article/10.3390/recycling7040058/s1. File 1. PET Bottle Waste Utilization.ahps; File 2. Questionnaire for Academician; File 3. Questionnaire for Stakeholders; File 4. Sub-Criteria Pairwise Comparison Matrix.

**Author Contributions:** Data collection and interpretation and writing—manuscript preparation, A.A.; writing—review, G.G.; supervision, C.I. All authors have read and agreed to the published version of the manuscript.

**Funding:** This research received no external funding.

**Data Availability Statement:** The data presented in this study are available on request from the corresponding author.

**Acknowledgments:** The authors acknowledge the support of the experts and stakeholders who provided valuable opinions to the questionnaire.

**Conflicts of Interest:** The research and its publication were prepared for academic purposes, and there is no relation to any external parties.

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
