# Peer review of "Analyzing Polyethylene Terephthalate Bottle Waste Technology Using an Analytic Hierarchy Process for Developing Countries: A Case Study from Indonesia"

_recycling, doi:10.3390/recycling7040058_

Round 1

Reviewer 1 Report

Comments and Suggestions

Amirudin et al. have studied a PET recycling strategy for developing countries. This manuscript looks like an investigation/technical report and literature mini-review, not a scientific research paper. But the reviewer thinks this report can fit some specific readers and recommends this manuscript to publish in Recycling after major revision. Some comments should be addressed as follows,

1.     The authors need to explain how to make figures 2-10,(where is the data from?) and they are hard to understand based on your methodology description. Please describe them in more detail.

2.     The reviewer suggested the authors can give the novelty and some scientific meanings of this manuscript.

3.     As we know, mechanical recycling is better for PET bottles. The authors can collect some new mechanical recycling methods to improve the quality of this manuscript.

4.     The Methods part should be described in more detail. Indeed, the research procedure could be much more clearly described by means of a bar diagram. the authors can compare the literature to emphasize the importance of this manuscript

5.     Conclusions can be improved and can extend to compare to Indonesia's government policy, which will increase the value of this manuscript.

Author Response

Thank you very much for your valuable review and correction. It will improve our paper and enrich the reader's understanding of PET bottle waste and environmental issues.

For detail revision and respond was presented below:

  1. The authors need to explain how to make figures 2-10,(where is the data from?) and they are hard to understand based on your methodology description. Please describe them in more detail.

We update the description in methodology part to improve the explanation how the figure 3-11 calculate in line 270-278, line 286-294 and presented detail calculation in supplementary material for each figure. We also convert the priority value from the eigenvector calculation into a percentage (%) to make the reading easier. Figure 3 is table 8, and figures 4-11 will be presented in detail in the supplementary material. We also attach the PET Bottle Waste Utilization.ahps file; this file is the original recap from expert choice version: 6.2.001.42753 to prove the originality and validity of the calculation. Finally, we try to show the technical procedure in the calculation in the supplementary material. Even though we believe it is redundant because equation (1) represents the equation in the expert choice application, we don’t calculate it manually to keep the process valid and precise.

  1. The reviewer suggested the authors can give the novelty and some scientific meanings of this manuscript.

We presented the novelty in several sessions, in abstract lines 17-21, background line 72-76, and line 81-88. We observed PET bottle waste technology comparison with the AHP method is a new preliminary study in this field. Our results are complementary to the findings of LCA studies.

  1. As we know, mechanical recycling is better for PET bottles. The authors can collect some new mechanical recycling methods to improve the quality of this manuscript.

It was not the intention of this work to demonstrate the superiority of mechanical recycling. Economic and environmental analyses are more appropriate for this. We were investigating the acceptance of technologies by stakeholders. Even well advanced technologies can fail to convince stakeholders, if they do not comply with the interests of important social groups. Nevertheless, we improved discussion lines 621-629 to show that pelletizing could be the initial step for further mechanical recycling improvement.

  1. The Methods part should be described in more detail. Indeed, the research procedure could be much more clearly described by means of a bar diagram. the authors can compare the literature to emphasize the importance of this manuscript.

We update the description in methodology part to improve the explanation in line 270-278 and line 286-294. We also update and presented the literature comparison in figure 1 in line 150-152 as summary of literature review and point out the novelty.

  1. Conclusions can be improved and can extend to compare to Indonesia's government policy, which will increase the value of this manuscript.

Was update in conclusion session line 664-670.

Please see the attachment to get the new version of the manuscript.

Reviewer 2 Report

The article describes an empirical study on waste management of polyethylene terephthalate (PET) bottles in Indonesia in which stakeholders from various fields including academic experts were surveyed. Several management options were compared according to pre-established criteria and sub-criteria using Analytical Hierarchy Process (AHP). The results suggest that pelletizing is an acceptable option because it is supported by social aspects creating jobs.

Comments:

The article is well structured, the Introduction provides a good framework for the work citing very recent references in the field of waste management and the methodology is clearly presented as well as the results. The authors justify the interest of the work saying that the Life Cycle Assessment widely used in waste management does not consider some aspects relevant to decision makers. LCA is a powerful tool for evaluating and comparing waste management alternatives, my main criticism of the work is that it does not use the results of LCA studies as a starting point. How do the authors propose to make their results compatible with the LCA results? Would it be possible, for example, to clearly inform all respondents about the preferable management options as a result of LCA and only then carry out the questionnaires? Without that, this work seems to me just “another study on PET recycling”.

Author Response

Comments and Suggestions for Authors

The article describes an empirical study on waste management of polyethylene terephthalate (PET) bottles in Indonesia in which stakeholders from various fields including academic experts were surveyed. Several management options were compared according to pre-established criteria and sub-criteria using Analytical Hierarchy Process (AHP). The results suggest that pelletizing is an acceptable option because it is supported by social aspects creating jobs.

Comments:

The article is well structured, the Introduction provides a good framework for the work citing very recent references in the field of waste management and the methodology is clearly presented as well as the results. The authors justify the interest of the work saying that the Life Cycle Assessment widely used in waste management does not consider some aspects relevant to decision makers. LCA is a powerful tool for evaluating and comparing waste management alternatives, my main criticism of the work is that it does not use the results of LCA studies as a starting point. How do the authors propose to make their results compatible with the LCA results? Would it be possible, for example, to clearly inform all respondents about the preferable management options as a result of LCA and only then carry out the questionnaires? Without that, this work seems to me just “another study on PET recycling”.

Thank you very much for your effort to improve our manuscript.

It is not necessary to make our results compatible with LCA. AHP is a complete different approach. The results of LCA are acknowledged by the selection of the experts, who are perfectly aware of the LCA results. By their knowledge, they decide the priorities of the alternatives (land filling, energetic, mechanical, chemical recycling) in dependence from the criteria (environment, society, economy, etc.). The stakeholder do not need to have a deep knowledge of the alternatives (including LCA results). They are not even asked about them. They only decide what is more important to them, i.e. environmental impact or social impact. Now, AHP provides a synthesis of the wishes of the stakeholders and the knowledge of the experts.

This provides an aid for decision making. Decisions are not made solely on the basis of LCA. LCA is based on process parameters that allow the calculation of environmental impacts in detail. In contrast, AHP is based on various experts' and stakeholders' perceptions, which are considered to have the understanding and ability to compare various criteria to choose the right technology. In this case, the results of our study complement the results of the LCA, so comparing our results with the LCA is not applicable because different aspects of the same problem are investigated. LCA uses indicators such as various types of pollution, climate change, and avoided resources, etc. However, these indicators may differ in their impact on the environment, making it difficult to obtain an ultimate conclusion. AHP provides such comparisons by asking stakeholders of their opinion. This includes also indicators that commonly not included in LCA.

Therefore, this is not a study about PET recycling (technology) at all. It is a study about finding a solution for treating PET waste that fits to the society that requires its implementation.

An additional passage was added accordingly (line 83-88).

Round 2

Reviewer 1 Report

The revised manuscript is now recommended for publication.

Author Response

thank you

This manuscript is a resubmission of an earlier submission. The following is a list of the peer review reports and author responses from that submission.

Round 1

Reviewer 1 Report

The main issue is the lack of scientific reference to literature. A strong effort is required to make the paper scientific.

Abstract has inappropriate structure. I suggest to answer the following aspects: - general context - novelty of the work - methodology used - main results 

Introduction presents interesting information. However, does not succeed to frame the framework within relevant literature. Scientific literature is almost absent. This section need to be reinforced: You could for example emphasize the role of circular economy also looking at social aspects. I would suggest to use this section to discuss about the relevance of waste materials for the application of circular economy principle. The circular economy approach has the goal to make better use of resources/materials through reuse, recycling and recovery, and also to minimise the energy and environmental impact of resource extraction and processing. Bio-based industry and bioenergy are paramount:

Please see:

https://doi.org/10.1016/j.erss.2021.102238

https://doi.org/10.1016/j.jclepro.2019.117868

https://doi.org/10.1016/j.scitotenv.2021.149605

https://doi.org/10.1016/j.enpol.2019.111220

https://doi.org/10.1016/j.renene.2020.12.034

The research methodology seems underdeveloped. Methods should be described in detail. Indeed, I think the research procedure could be much more clearly described by means of a diagram also highlighting its potential and limit.

Results need to be discussed in light of literature.

 Conclusions are extremely succinct. I suggest to authors to propose policy directions in a broader sense. Link with future lines of research should look at financial sustainability of projects. A clear example is “green finance”.

Reviewer 2 Report

The paper "Analyzing Polyethylene Terephthalate Bottle Waste Technology Using Analytic Hierarchy Process for Developing Countries: A case Study from Indonesia falls in topics of journal "Sustainability". Also the paper according to my opinion doesn't meet the standard quality of the paper that should be published in one prestigious journal as "Sustainability". A lot of core elements of one well-written and performed study are missing. Therefore, my conclusion is that the paper should be rejected. An explanation of my decision can be found below.
1) Clearly described aims, the main contributions, novelty, and verification of results are missing in the paper. 
2) The overall structure of the paper and quality is very poor.
3) In the introduction section you gave an overview of other relevant studies, and try to show the link between previous studies and your paper. That is good, but the following tasks should be fulfilled also:
the introduction should give an overview of the field significance, and should consider the following main questions: What are the gaps in literature? What are the contributions of this study? What are the main aims of this article?"
4) Most of the current section Introduction should be moved to new-formed 2. Literature review.
5) In your paper, you made effort to show novelty, contributions, but you didn't convince me in your approach. 
6) What new brings your paper? You have only written well-known facts which have been published before from the aspect of the methodology. The application of the AHP method is not new. Has been published in many papers. This method is old and in some cases not provide good results. You should use some of the newer methods like SWARA, FUCOM, BWM, or make integration with some of the uncertainty theories (fuzzy, rough, neutrosophic)
7) Hierarchy model in the paper isn't well-formed. There are 6 main criteria. The first has 6, while the second has 2 sub-criteria, etc. This means that all further calculation isn't good, so the model isn't good also. Please carefully read this. We have a limitation SUMwj=1.00 for both levels of the hierarchy. What is happening? It is obvious that sub-criteria that belongs to the 2nd main criterion have greater values than sub-criteria that belong to the 1st criterion... They don't deserve that. If we consider the following situation. Two criteria have an equal value of 0.50 and each sub-criteria have an equal value situation will be: sub-criteria belonging to C1 will have values of 0.083, while sub-criteria belonging to C2 will have values of 0.250. This isn't realistic, so your results are not correct. A hierarchy structure with three levels must have an approximately equal number of sub-criteria for each main criteria.
- Sensitivity analysis is missing.
- Comparative analysis with other MCDM methods is missing.
- Quality comparison analysis is missing. Why didn't compare your approach with other existing methods and ensure discussion?
- Conclusion is very poor.

Reviewer 3 Report

Dear authors,

Thank you for your effort in addressing the sustainability issue in Indonesia. 

This paper analyzed the preferable waste treatment option for PET bottles in Indonesia using the AHP method. The study provided a strong literature review and collected a reasonable size of stakeholders' and experts' opinions. Overall the quality of study is good. 

Below are my comments.

1. I find it difficult to understand and/or reproduce the numerical results (figure 2 to 6) based on your methodology description. Please consider elaborating on them. 

  1. Figure 2-5. What questions have been asked to stakeholders to yield the results; How to convert the raw numerical results to the final percentage representations?

  2. Attachment of the original questionnaire can be helpful.

  3. Figure 6. Are the results weighted based on the priority determined in Figure 2? If so, how is it calculated? Line 193, you implied a weighting process.  

  4. Figure 6. Add a legend for the percentage on the y-axis, or in the figure caption.

2. Check the consistency of format for decimal points (i.e., sometimes comma, sometimes full stop).

3. Add a line or two in discussion to comment on the representativeness of selected stakeholders for this study. Because the results can be influenced by their subjective opinion.

4. Consider splitting the discussion section into two sub-sections, around Line 438, to improve reading friendliness. 

Best regards,